# Developing a Dual-Stream Deep-Learning Neural Network Model for Improving County-Level Winter Wheat Yield Estimates in China

Hai Huang [1], Jianxi Huang [1,2,*], Quanlong Feng [1,2], Junming Liu [1,2], Xuecao Li [1,2], Xinlei Wang [1] and Quandi Niu [1]

1 College of Land Science and Technology, China Agricultural University, Beijing 100083, China
2 Key Laboratory of Remote Sensing for Agri-Hazards, Ministry of Agriculture and Rural Affairs, Beijing 100083, China
* Correspondence: jxhuang@cau.edu.cn

**Abstract:** Accurate and timely crop yield prediction over large spatial regions is critical to national food security and sustainable agricultural development. However, designing a robust model for crop yield prediction over a large spatial region remains challenging due to inadequate surveyed samples and an under-development of deep-learning frameworks. To tackle this issue, we integrated multi-source (remote sensing, weather, and soil properties) data into a dual-stream deep-learning neural network model for winter wheat in China's major planting regions. The model consists of two branches for robust feature learning: one for sequential data (remote sensing and weather series data) and the other for statical data (soil properties). The extracted features by both branches were aggregated through an adaptive fusion model to forecast the final wheat yield. We trained and tested the model by using official county-level statistics of historical winter wheat yields. The model achieved an average $R^2$ of 0.79 and a root-mean-square error of 650.21 kg/ha, superior to the compared methods and outperforming traditional machine-learning methods. The dual-stream deep-learning neural network model provided decent in-season yield prediction, with an error of about 13% compared to official statistics about two months before harvest. By effectively extracting and aggregating features from multi-source datasets, the new approach provides a practical approach to predicting winter wheat yields at the county scale over large spatial regions.

**Keywords:** winter wheat; crop yield prediction; deep learning; remote sensing; weather data; soil data




## 1. Introduction

Accurate and timely crop yield prediction is critical for ensuring food security and planning production, storage, transportation, and other interconnected activities [1]. The increasing challenges to agriculture, including climate change, a growing population, and the degradation of cultivated land require effective methods to support a global management system for sustainable agricultural development [2]. As the world's most populous and largest developing country, China has been devoted to achieving a high self-sufficiency for cereal crops. With China's aging farming population, accelerating urbanization, and growing demand for a high-protein diet, food security will continue to be the country's top priority [3].

Remote sensing offers the benefits of a synoptic view, multi-temporal coverage, easy access, and cost-effectiveness and is, therefore, a promising approach to crop yield prediction [4–6]. The strategies for predicting crop yields with remote-sensing data can be divided into two main categories: statistical and process-based models. The former usually assumes that the photosynthetic capacity of crops, which can be inferred by surface reflectance or spectral vegetation indices, is directly related to crop yield [7]. Several studies have been based on statistical regression for the relationship between crop yields and variables such as

remote-sensing indexes [8–12], reflectance [13,14], or backscattering [15]. Machine-learning methods such as support vector machines and random forest models have been used to predict crop yield [16–18]. These machine-learning methods could potentially offer great performance for non-linear relationships. However, they are mainly based on spectral features such as the normalized difference vegetation index and may fail to extract high-level representative information that could be more closely related to crop yields.

The second category, process-based models, simulates crop yield dynamically using a well-calibrated crop growth model, in which remote-sensing data is usually used to reinitialize or recalibrate the model or update its state variables at a finer spatial resolution than the driving data [19–27]. This strategy provides a mechanical and explanatory model for crop yield components. Nevertheless, the calibration of process-based models across large spatial regions remains challenging. It requires numerous field measurements, such as soil and crop characteristics and field management practices (e.g., irrigation, fertilization).

Deep learning is capable of modeling high-level feature representations through a hierarchical learning framework [28], and has achieved surprisingly successful performance in many computer-vision tasks, such as image classification [29], object detection [30,31], and semantic segmentation [32]. The most widely used deep-learning models include convolutional and recurrent neural networks. Among the convolutional neural networks, AlexNet [29] confirms the superior ability of deep-learning neural networks over proxies such as manual indexes to discriminate between and represent critical features. Many deeper and larger network structures have been developed, such as GoogLeNet (also known as Inception-v1) [33], VGG [34], ResNet [35], and DenseNet [36], and have strengthened the deep-learning ability of models. Among the recurrent neural networks, the long short-term memory (LSTM) model [37] and the gated recurrent unit (GRU) model [38] are widely used because they are good at modeling the temporal dependencies of sequential signals and show superior performance compared with other recurrent neural networks in finding long-term relationships.

Deep learning is becoming increasingly popular in remote sensing [39–44], and researchers have started employing deep-learning models based on remote-sensing data for crop yield prediction [45–50]. Previous studies have adopted deep-learning methods to predict corn yields at a county level in the U.S. [16,51]. They used MODIS remote-sensing and weather data to achieve a correlation coefficient of about 0.8 ($R^2$ = 0.64) between the predicted and official statistical corn yields. However, their models only cascaded several fully connected layers with no modern deep-learning structures such as convolutional neural networks or recurrent neural networks, thereby limiting the model's performance. You et al. [45] proposed a deep Gaussian process model to estimate soybean yields in the USA using only MODIS data and achieved an average $RMSE$ = 373 kg/ha and $MAPE$ = 15% compared with USDA predictions from 2011 to 2015. You et al. [45] used convolutional and recurrent-neural-network methods. The convolutional neural network performed better, with the average $RMSE$ dropping by 8%, which agrees with our findings. As a representative structure of recurrent neural networks, LSTM has also been utilized by Wang et al. [52] for soybean yield prediction with maximum $R^2$ = 0.58 and $RMSE$ from 230 to 700 kg/ha. Oliveira et al. [46] achieved good performance for US soybean (with $R^2$ = 0.75 and $RMSE$ = 354 kg/ha) and maize (with $R^2$ = 0.71 and $RMSE$ = 1393 kg/ha) yield prediction with LSTM. Auto-encoder has also been studied by Ma et al. [47] to estimate rice yields in South Korea based on remote-sensing and weather data and achieved a better performance ($RMSE$% = 6.89%) than traditional artificial neural networks ($RMSE$% = 8.03%). However, none of the abovementioned studies simultaneously considered remote-sensing, weather, and soil data, which fails to take complete advantage of multi-source data to provide a more comprehensive characterization of the crop growth process and improve the accuracy of the prediction model. The extended LSTM is always used in predicting crop yields by multi-source data, such as rice yield prediction [53], smallholder maize yield estimation [54] in China, and corn yield estimation in the USA corn belt [50]. Luo et al. [55] developed an LSTM-based framework, mapping the global wheat production

with $R^2 = 0.76$ in the USA. Tian et al. [56] estimated wheat yields by remote-sensing and meteorological data, achieving better performance in the Guanzhong Plain ($R^2 = 0.83$ and *RMSE* = 357.77 kg/ha). These methods have tried to extract more discriminative yield-related features by combining multi-source data. However, these methods are weak in exploring the feature-learning structure. In addition, few previous have studies compared the suitability of the convolutional neural network and recurrent neural network structures for crop yield prediction.

In the present study, we developed a novel dual-stream deep-learning neural network model to improve regional prediction of crop yields by integrating remote-sensing data with other feature data (specifically weather and soil data). Our objectives were to (i) compare the capabilities of well-known convolutional neural networks and recurrent neural networks for winter wheat yield prediction in China, (ii) develop a novel adaptive deep-fusion model that could integrate multi-source data to improve yield prediction, and (iii) assess the model's prediction performance for the major winter wheat planting areas of China.

## 2. Study Area and Dataset

### 2.1. Study Area

This study focused on China's major winter wheat growing regions (Figure 1), which account for about 95% of China's total winter wheat planting area. This area extends from 98°16′E to 122°43′E and from 22°27′N to 40°56′N, which represents a vast geographical area. It includes two main winter wheat planting regions: the northern and southern areas. The northern area lies south of the Great Wall and north of the Qinling Mountains and Huaihe River, accounting for more than half of China's wheat planting area. It has a temperate continental monsoon climate, with an annual average temperature of 9 to 15 °C and a yearly rainfall of 440 to 980 mm. The primary prevailing crop rotation is winter wheat, followed by summer corn. The southern area is located south of the Qinling Mountains and Huaihe River. It has a subtropical monsoon climate, with an annual average temperature of 16 to 24 °C and annual precipitation of more than 1000 mm. Winter wheat is mostly rotated with paddy rice in this area. The study area comprises 14 provinces and two provincial-scale municipalities: Beijing, Tianjin, Hebei, Shanxi, Jiangsu, Anhui, Shandong, Henan, Hubei, Chongqing, Sichuan, Guizhou, Yunnan, Shaanxi, Gansu, and Ningxia. This area includes a total of average 555 winter wheat planting counties (Figure 1). Winter wheat in the study area is usually sown from September to late October and harvested from late May to mid-June of the following year. Its phenological development is closely related to latitude, that is, the development stage in the south is usually earlier than in the north.

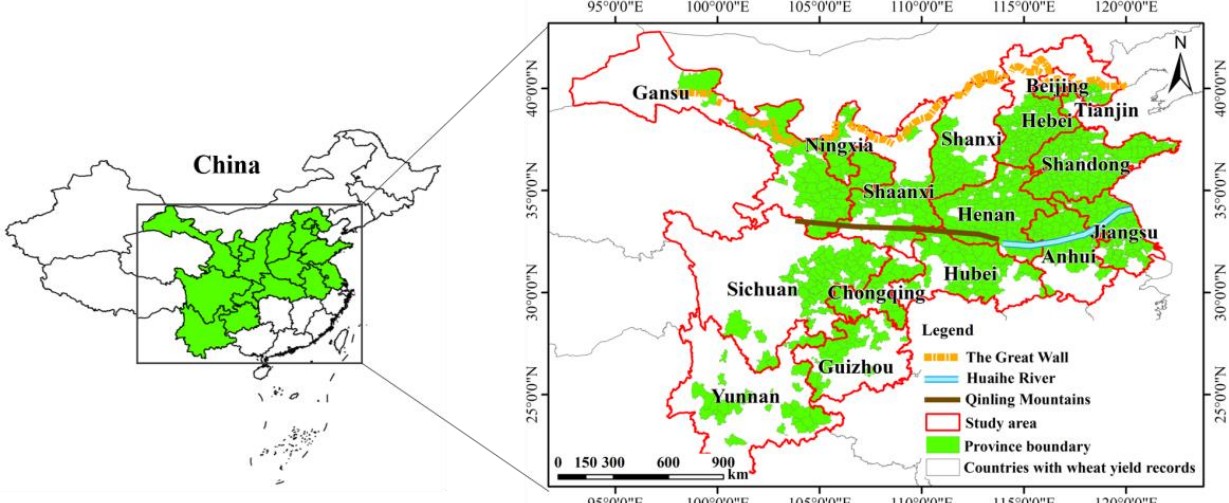

**Figure 1.** The study area of the counties where winter wheat is planted in China.

The geospatial vector data of administrative divisions shown in Figure 1 was sourced from the National Catalogue Service for Geographic Information (https://www.webmap.cn/commres.do?method=result100W, accessed on 1 September 2019).

### 2.2. Dataset

Crop yield is affected by many factors, including weather conditions, soil properties, irrigation systems, cultivation techniques, fertilizers, and crop variety [47,57–61]. Integrating all of these factors within a yield-prediction model is challenging because some specific datasets are difficult to obtain. In the present study, we used three major data sources to account for the yield predictor indicators: remote-sensing, weather, and soil property data.

### 2.2.1. Remote-Sensing Data

MODIS-derived products from the National Aeronautics and Space Administration (NASA) have been widely used in crop yield prediction due to their free, open access, and global coverage [7,57,62]. Among these products, we used MOD09A1 (Terra 8-day 500 m surface reflectance, https://lpdaac.usgs.gov/products/mod09a1v006/, accessed on 1 September 2019) and MOD11A2 (Terra 8-day 1 km land surface temperature/emissivity, https://lpdaac.usgs.gov/products/mod11a2v006/, accessed on 1 September 2019) within a growing-season from 2001 to 2015, which provide an 8-day composite value for seven surface spectral reflectance bands and two temperature bands (daytime and nighttime land surface temperature).

### 2.2.2. Weather Data

We obtained the weather data used in this study from the China meteorological forcing dataset, which is maintained by the Institute of Tibetan Plateau Research, Chinese Academy of Sciences [63,64]. The dataset provides seven gridded meteorological variables, including temperature, air pressure, specific humidity, wind speed, downward shortwave radiation, downward longwave radiation, and precipitation rate. The spatial resolution is 0.1°, and the temporal resolution is 3 h [65].

### 2.2.3. Soil Property Data

Soil property data were downloaded from SoilGrids.org, which provides data on nine features that describe soil physical and chemical properties for most of the world, including clay, silt, sand content, coarse fragments, and bulk density [46,66]. These variables are available at seven depths from 0 to 200 cm with a spatial resolution of 1 km or 250 m [65].

### 2.2.4. Cropland Land Cover Data

A crop mask for winter wheat is needed to delineate winter wheat's spatial distribution in a yield-prediction model. In this study, we used the MODIS Land Cover Product because winter wheat is the dominant crop in the cropland during the growing season in China. Although this could introduce certain errors, previous studies [52,67] justified using the MODIS Land Cover Product for crop yield prediction because they found that this approach did not degrade the model results compared with higher resolution crop-type maps. Therefore, the MODIS-derived MCD12Q1 product was used in this study, which provides annual global land cover data at a resolution of 500 m, to distinguish cropland from non-cropland.

### 2.2.5. County-Level Yield Data

We obtained winter wheat yield data of an average of 555 counties from China's county-level statistical yearbooks from 2001 to 2015, except for 2006 because of missing data. All yields were reported as kg/ha. Table 1 summarizes the input variables used in the deep-learning model.

**Table 1.** Input variables are used in the deep-learning model.

| Data Source | Variable Name | Units of Measurement | Spatial Resolution | Temporal Resolution | Description |
|---|---|---|---|---|---|
| Remote-sensing data | sur_refl_b01 | | 500 m | 8 d | Surface reflectance band 1 (620–670 nm) |
| | sur_refl_b02 | | 500 m | 8 d | Surface reflectance band 2 (841–876 nm) |
| | sur_refl_b03 | | 500 m | 8 d | Surface reflectance band 3 (459–479 nm) |
| | sur_refl_b04 | | 500 m | 8 d | Surface reflectance band 4 (545–565 nm) |
| | sur_refl_b05 | | 500 m | 8 d | Surface reflectance band 5 (1230–1250 nm) |
| | sur_refl_b06 | | 500 m | 8 d | Surface reflectance band 6 (1628–1652 nm) |
| | sur_refl_b07 | | 500 m | 8 d | Surface reflectance band 7 (2105–2155 nm) |
| | LST_Day | | 1 km | 8 d | Daytime land surface temperature |
| | LST_Night | | 1 km | 8 d | Nighttime land surface temperature |
| Weather data | temp | K | 0.1 | daily | Instantaneous near-surface (2 m) air temperature |
| | pres | Pa | 0.1 | daily | Instantaneous near-surface (2 m) air pressure |
| | shum | $kg\,kg^{-1}$ | 0.1 | daily | Instantaneous near surface (2 m) air specific humidity |
| | wind | $m\,s^{-1}$ | 0.1 | daily | Instantaneous near-surface (10 m) wind speed |
| | srad | $W\,m^{-2}$ | 0.1 | daily | Surface downward shortwave radiation |
| | lrad | $W\,m^{-2}$ | 0.1 | daily | Surface downward longwave radiation |
| | prec | $mm\,hr^{-1}$ | 0.1 | daily | Precipitation rate |
| Soil data | BLDFIE | $kg\,m^{-3}$ | 1 km | | Bulk density (fine earth) |
| | CECSOL | $cmolc\,kg^{-1}$ | 1 km | | Cation exchange capacity of the soil |
| | CLYPPT | % | 1 km | | Clay content (0 to 2 μm) mass fraction |
| | CRFVOL | % | 1 km | | Coarse fragment volumetric fraction |
| | ORCDRC | $g\,kg^{-1}$ | 1 km | | Soil organic carbon content (fine earth fraction) |
| | PHIHOX | | 1 km | | pH × 10 in $H_2O$ |
| | PHIKCL | | 1 km | | pH × 10 in KCl |
| | SLTPPT | % | 1 km | | Silt content (2 to 50 μm) mass fraction |
| | SNDPPT | % | 1 km | | Sand content (50 to 2000 μm) mass fraction |

## 3. Methods

We integrated the multi-source remote-sensing, weather, and soil property data into a deep-learning model for winter wheat yield prediction in China. The multi-source data could be categorized into two types: dynamic datasets (including remote-sensing and weather data) and static datasets (the soil properties). Therefore, a dual-stream deep-learning neural network was developed between the crop yield and the aggregated features extracted and integrated from these two data types. Figure 2 illustrates the framework for this analysis, which consisted of six main components: (a) data preprocessing; (b) the remote-sensing–weather branch; (c) the soil branch; (d) the fusion module; (e) network training; and (f) accuracy assessment. Details of these components are provided in the following sections.

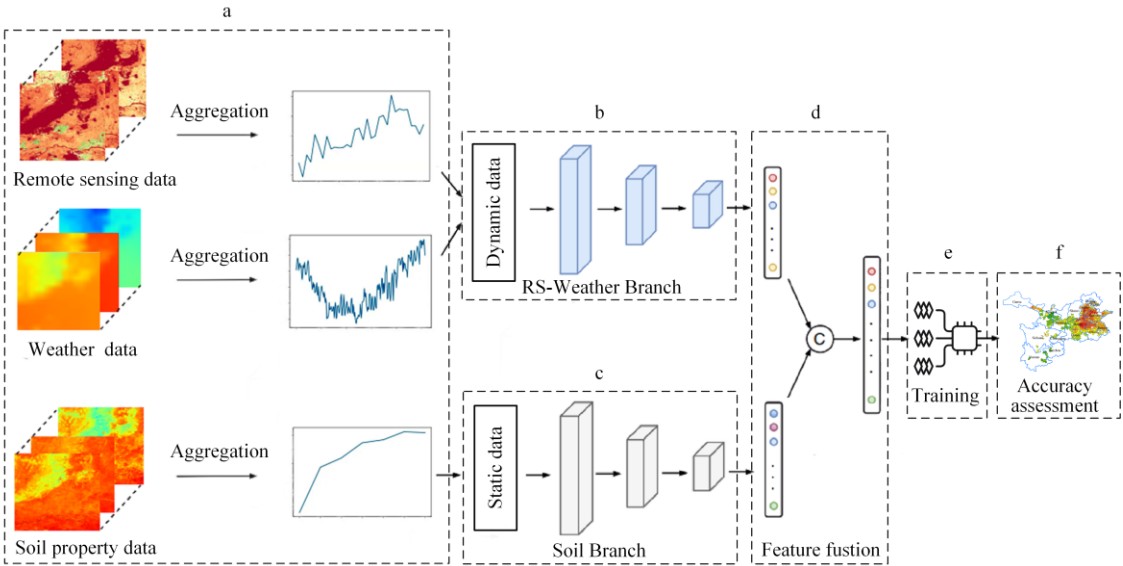

**Figure 2.** An overview of the proposed crop yield prediction method. (**a**) Preprocessing for the multi-source data. (**b**) The dynamic stream comprised the remote-sensing–weather (RS–weather) branch. (**c**) The static stream comprised the soil branch. (**d**) The fusion module integrated the dynamic and static streams. (**e**) Network training. (**f**) Crop yield prediction and accuracy assessment.

### 3.1. Data Preprocessing

The weather variables were first resampled to an 8-day temporal resolution using the mean-value composite method to be consistent with the MODIS 8-day product data. We then aggregated all remote-sensing, weather, and soil property data at a county level using the average value after being masked to the cropping area within the county. We then used min–max normalization to convert all data to values between 0 and 1 because the remote-sensing and weather data were continuous sequential variables. In contrast, soil properties remain almost constant over time. We compiled the first two datasets as dynamic data and the soil dataset as static data [46]. The period for the dynamic data was from October to mid-June of the following year, corresponding to winter wheat's sowing and maturity stages. The remote-sensing and weather data were combined into a $16 \times 32$ matrix that comprised nine remotely sensed variables plus seven weather variables for 32 eight-day periods. Soil variables were compiled into a two-dimensional array with seven depths and nine soil properties and provided the model's static inputs.

### 3.2. Remote-Sensing–Weather Branch

An accurate winter wheat yield prediction requires extracting representative features from the input data. We combined the time series of remote-sensing images and weather data throughout the winter wheat growing season to generate the sequential dataset for feature extraction in the RS–weather branch. To compare the performance of several well-known deep-learning models for yield prediction, we modeled the network structure of the RS–weather branch using four convolutional neural network models: VGG [68], Inception [33], ResNet [35], and DenseNet [36] We also compared two recurrent neural network models: the long short-term memory (LSTM) model [37] and the gated recurrent unit (GRU) model [38]. All the network variants of the RS–weather branch are shown in Figure 3. Our experimental results (see the Section 4 for details) indicated that the Inception-based model yielded the highest accuracy, so we selected this model for subsequent analysis of the RS–weather branch (Figure 4).

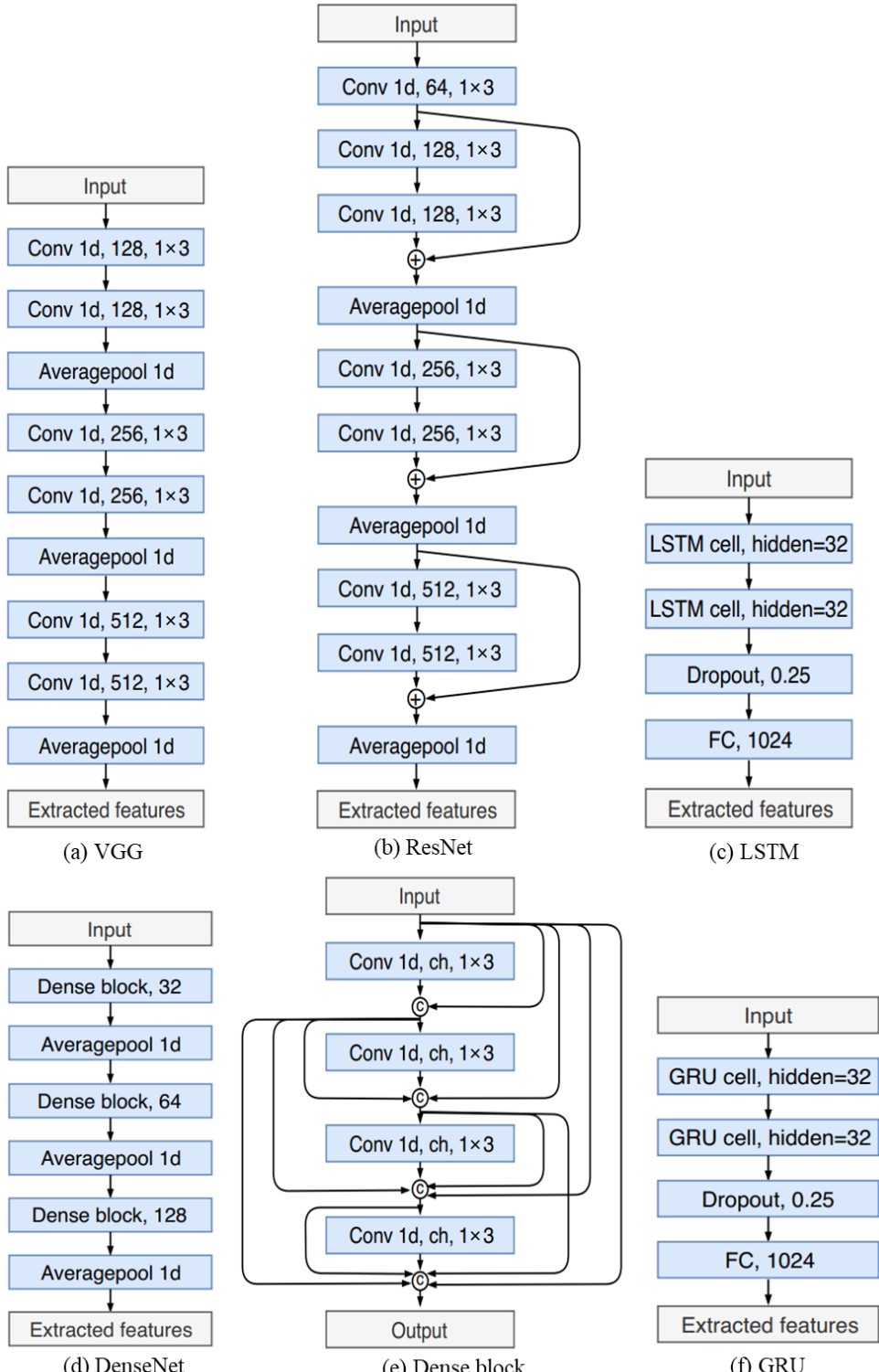

**Figure 3.** Network variants for the RS–weather branch: (**a**) VGG, (**b**) ResNet, (**c**) long short-term memory (LSTM) (**d**) DenseNet, (**e**) Dense block, and (**f**) gated recurrent unit (GRU).

As depicted in Figure 4, we used only one-dimensional convolutions (Conv 1 d) and average pooling (Average pool 1 d). To increase our ability to generalize the original Inception modules, we added a dropout layer [69] with a rate of 0.25 after each convolution layer. One of the Inception module's merits is that it can extract features from a multi-scale receptive field [33], resulting in a strong ability to learn short- and long-range dependencies. The network variants for the RS–weather branch are illustrated as follows.

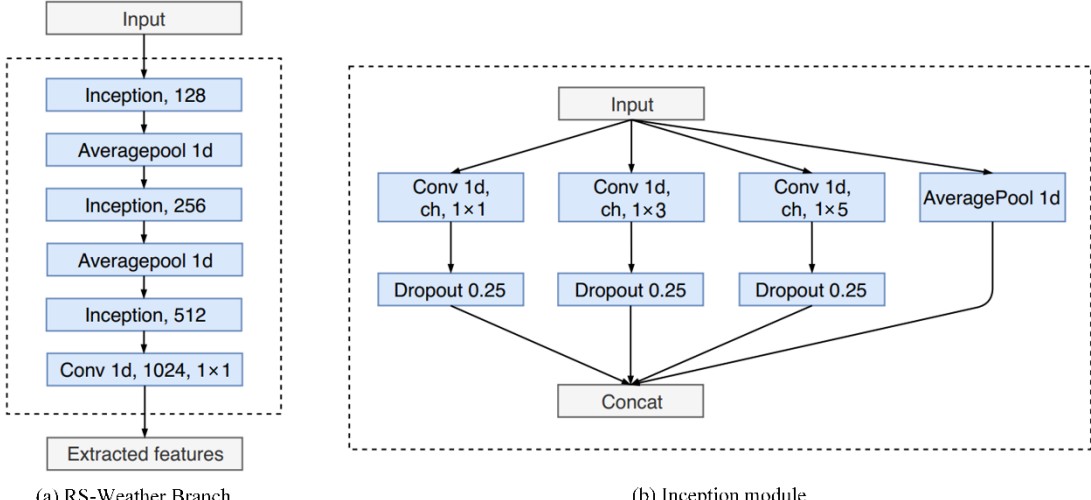

**Figure 4.** The architecture of the RS–weather branch: (**a**) the RS–weather branch; (**b**) the Inception module.

### 3.3. Soil Branch

Previous studies [46] have demonstrated that soil property data are vital for accurate crop yield prediction. Therefore, incorporating soil property data provides both unique information (different from the sequential data) and complementary information and could increase yield-prediction performance. Given that the soil property data are relatively stable compared to the RS–weather sequential data, we separated the soil branch from the RS–weather branch. Figure 5 shows the architecture of the soil branch. Only one-dimensional convolutions were employed to decrease the model's complexity, reducing the risk of overfitting the input data [42].

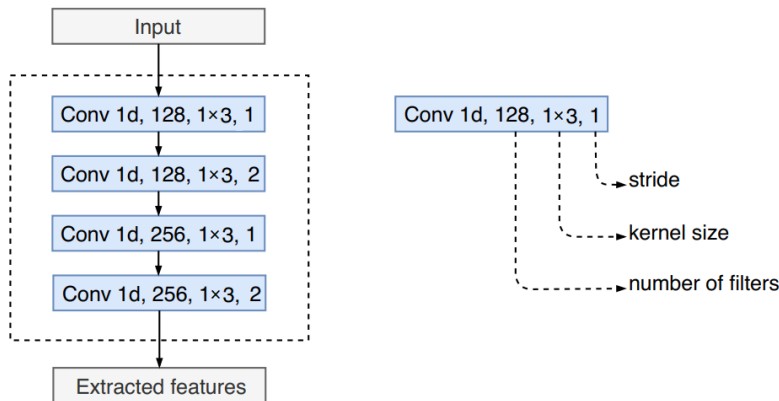

**Figure 5.** The architecture of the soil branch.

### 3.4. Fusion Module

After extracting the features of the multi-source data from the RS–weather data and the soil property data, we conducted feature-level fusion to provide the final yield prediction. Previous studies [46] simply stacked and concatenated these features without considering the relationships among them and their relative importance for yield prediction. In the present study, inspired by both the Squeeze-and-Excitation Network (SENet) model [70] and our previous research [42], we proposed an adaptive fusion module to aggregate the features from the RS–weather branch and the soil branch.

Figure 6 depicts the structure of the feature-fusion module that we used to recalibrate or re-weight the input features. The weights assigned to the elements can be automatically learned end-to-end. Specifically, we generated a channel descriptor after the input feature

was passed through a global average pooling layer [70]. Next, the channel-wise weight was learned by two consecutive fully connected layers and a sigmoid layer [70]. After feature fusion, the informative features were enhanced. In contrast, the less valuable features and noise were suppressed [42], which could effectively improve the representativeness and robustness of the fused features.

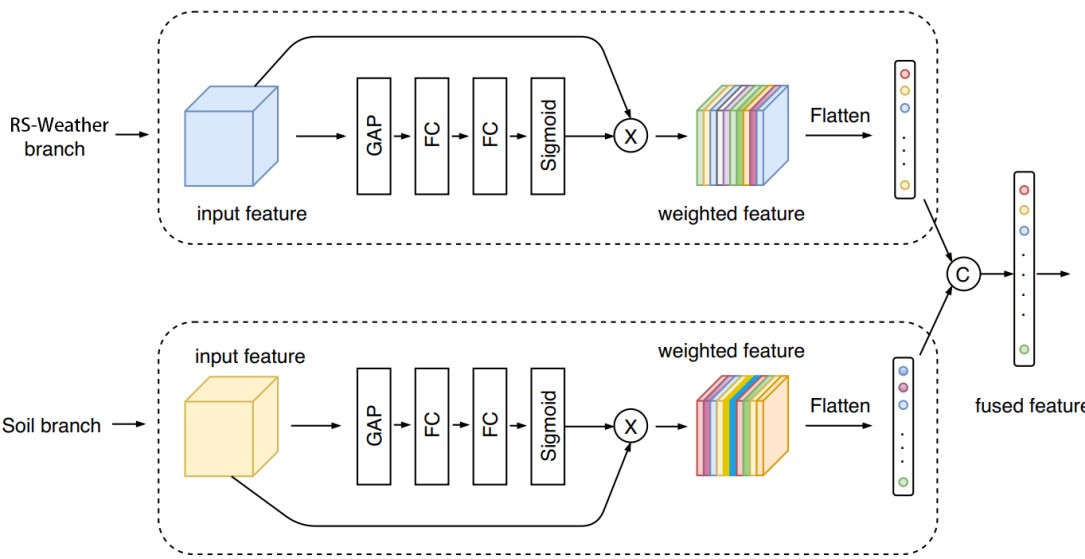

**Figure 6.** Structure of the data fusion module.

### 3.5. Network Training

Since the yield prediction model was trained from scratch (i.e., no pre-trained model was used), it was necessary to determine how to initialize all the model parameters. In this study, we initialized the model weights using the normalization method of [71], but the initial biases were set to zero.

During the network training, we used the Adam optimization method [72] due to its ability to adjust the learning rate automatically, leading to a faster and more stable training procedure. The initial learning rate for Adam was set to $10^{-3}$. We adopted the early-stop strategy to select the best model with the minimum validation loss.

Since the winter wheat yield prediction is a regression problem, we calculated the L2 loss based on the mean-squared error (*MSE*) [45].

$$MSE = \frac{1}{n} \sum_{i=1}^{n} (y_i - y_i^p)^2 \tag{1}$$

where $y_i$ and $y_i^p$ represent the official statistical and predicted winter wheat yields for county *i*, respectively, and *n* is the number of counties.

In this study, 90% of the official statistical data were randomly chosen as training samples to optimize the yield prediction model. The remaining 10% of the data was used as the validation dataset to evaluate the performance of the training. Since all the training and validation datasets were randomly selected, the model could learn from various soil, weather, and growth status scenarios. The proposed winter wheat yield prediction model was trained with the TensorFlow library (https://tensorflow.google.cn/, accessed on 1 September 2019).

### 3.6. Accuracy Assessment

To quantify the effectiveness of the proposed model for winter wheat yield prediction, we calculated the coefficient of determination ($R^2$), root-mean-square error (*RMSE*), mean absolute percentage error (*MAPE*), and mean error (*ME*):

$$R^2 = 1 - \sum_{i=1}^{n} \left(y_i^p - y_i\right)^2 / \sum_{i=1}^{n} \left(y_i - \overline{y}\right)^2 \qquad (2)$$

$$RMSE = \sqrt{\frac{1}{n}\sum_{i=1}^{n}\left(y_i - y_i^p\right)^2} \qquad (3)$$

$$MAPE = \frac{100}{n}\sum_{i=1}^{n}\left|\frac{y_i^p - y_i}{y_i}\right| \qquad (4)$$

$$ME = \frac{1}{n}\sum_{i=1}^{n}\left(y_i^p - y_i\right) \qquad (5)$$

where $\overline{y}$ is the mean value of the official statistical yield, and $y_i$ and $y_i^p$ have the same meaning as in Equation (1). An effective yield prediction model should simultaneously have a high $R^2$, low $RMSE$, low $MAPE$, and low $ME$. We also used leave-one-out cross-validation [47,67].

## 4. Results

### 4.1. Comparing Different Deep-Learning Models to Predict Winter Wheat Yield

Table 2 indicates that the Inception-based model had the highest accuracy for predicting China's winter wheat yield from 2001 to 2015, with a mean $R^2$ of 0.79, an $RMSE$ of 650.21 kg/ha, and a $MAPE$ of 12.4%. The ResNet-based model had the second-best performance, with a slightly lower $R^2$ (0.78) and a higher $RMSE$ (660.34 kg/ha) but a lower $MAPE$ (12.2%) than the Inception-based model. The LSTM and DenseNet models performed similarly, whereas VGG and GRU had inferior accuracy. Therefore, we selected the Inception-based model for the RS–weather branch of the proposed dual-stream deep-learning neural network.

**Table 2.** Performance of different deep-learning models based on the leave-one-out cross-validation. All methods produced statistically significant results ($p < 0.05$). The model with the best fit is boldfaced. *MAPE*, mean absolute percentage error; *RMSE*, root-mean-square error.

| Method | $R^2$ | RMSE (kg/ha) | MAPE (%) | ME (kg/ha) |
|---|---|---|---|---|
| VGG | 0.76 | 692.39 | 13.16 | 129.55 |
| ResNet | 0.78 | 660.34 | 12.18 | 55.83 |
| DenseNet | 0.78 | 663.87 | 12.60 | 79.72 |
| Inception | 0.79 | 650.21 | 12.37 | 54.21 |
| LSTM | 0.78 | 678.25 | 12.95 | 17.14 |
| GRU | 0.76 | 704.45 | 13.40 | 22.13 |

Figure 7 shows the *RMSE* of the deep-learning models for each year between 2001 and 2015. The Inception-based model had a lower *RMSE* than other models in most years and achieved the best accuracy in 2011, with the lowest *RMSE* (506.21 kg/ha). The models based on recurrent neural networks (LSTM and GRU) had higher accuracy than models based on convolutional neural networks from 2001 to 2003.

Our results indicated the average winter wheat yield statistics and the prediction results from the six deep-learning models from 2001 to 2015. From 2001 to 2003, most of the deep-learning models overestimated the winter wheat yield. According to historical climate data and previous studies [73], from 2001 to 2003, there was a severe drought in China, leading to significant differences in yields compared with the other years and resulting in the low performance of most of the deep-learning models. However, the yield predictions by LSTM and GRU were closer to the actual values during these years than models based on convolutional neural networks, which suggests that it could be possible to use these models to replace the convolution models in years with severe drought (Figure 8).

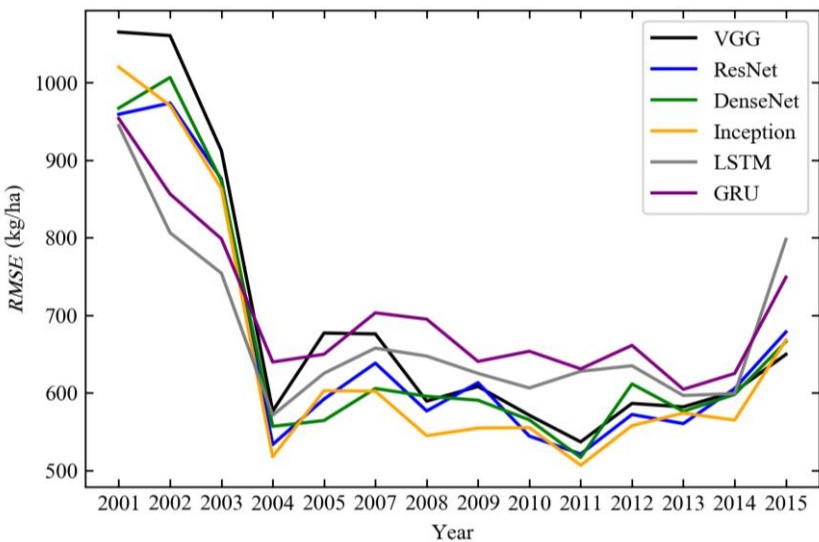

**Figure 7.** Root-mean-square errors (*RMSE*) of each model from 2001 to 2015.

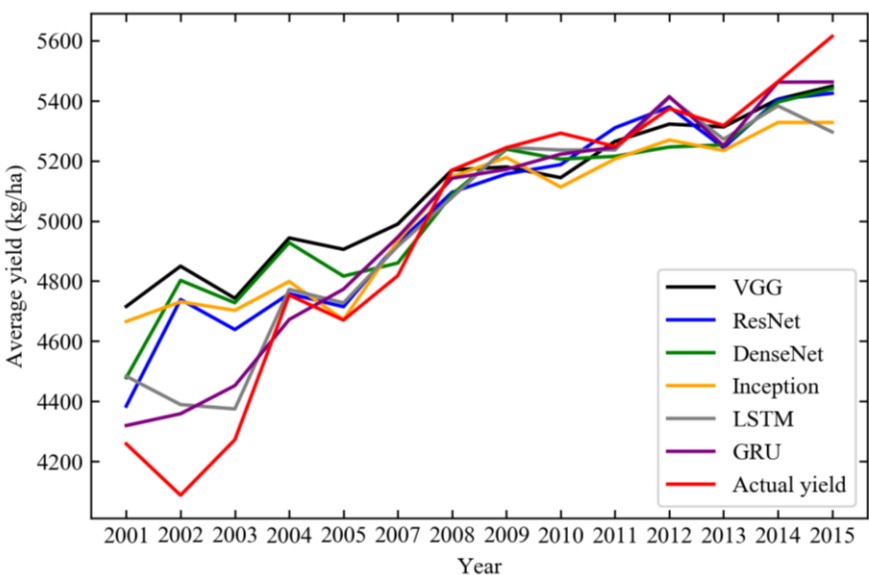

**Figure 8.** Comparison of predicted average yield and official statistical yield from 2001 to 2015.

### 4.2. Spatial Variation in Winter Wheat Yield Predictions

As mentioned in Section 4.1, the Inception-based model provided the best accuracy (the highest mean $R^2$ and lowest *RMSE*) for the period from 2001 to 2015, with the lowest *RMSE* (506 kg/ha) in 2011. Therefore, we analyzed the spatial pattern of yield prediction results at the county level in 2011 using the results of the Inception-based model (Figure 9).

Figure 9a shows the official statistical winter wheat yield in 2011, and Figure 9b shows the corresponding results predicted by the dual-stream deep-learning neural network model. The two figures show similar patterns, indicating that the developed deep-learning model performed well nationally. Figure 9c shows the spatial distribution of relative error for the proposed model. The relative error was less than 15% for 83% of the counties in the North China Plain, although several counties in northwestern and southwestern China had a high relative error (>25%). With our new model, the $R^2$ and *RMSE* in 2011 were 0.89 and 506 kg/ha, respectively, which shows high prediction accuracy. The results showed no distinct overestimation or underestimation, indicating that the dual-stream deep-learning neural network model provided unbiased estimates.

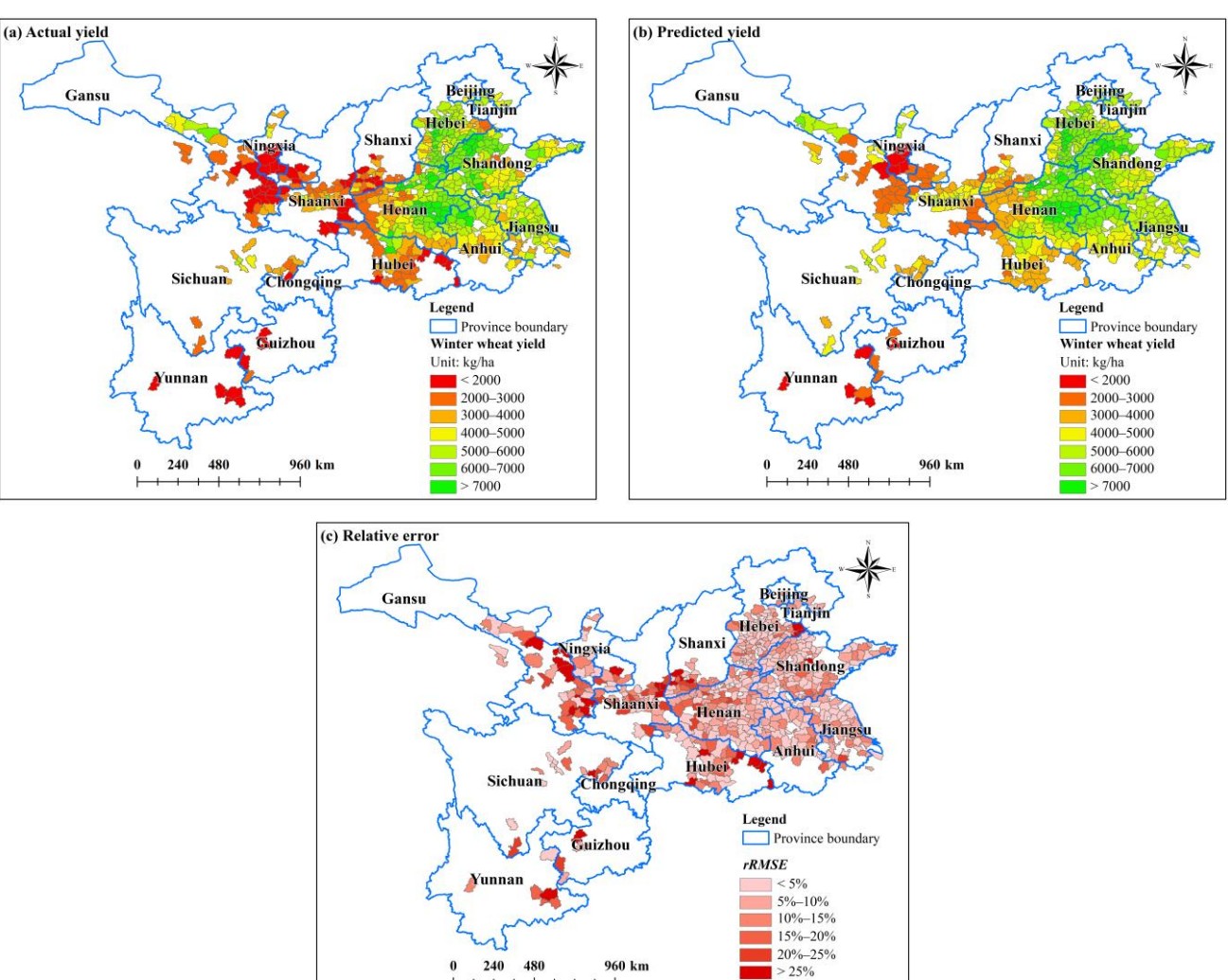

**Figure 9.** Statistical and predicted winter wheat yield in 2011. (**a**) The official statistical yield. (**b**) The predicted yield. (**c**) The relative error.

## 5. Discussion

### 5.1. Impact of Different Data Sources on Winter Wheat Yield Prediction

Due to the multi-source data (i.e., remote-sensing, weather, and soil property data) being highly correlated with winter wheat yield, ablation analysis [74] can be used to evaluate each data source's impact on yield prediction performance. Therefore, we performed a series of experiments to establish the yield-prediction model using individual remote-sensing data, weather data, or soil property data. The Inception prediction model was used, and leave-one-out cross-validation was used to assess the model's accuracy.

Our results showed that remote-sensing data alone achieved the highest prediction accuracy, with a mean $R^2$ of 0.73, an *RMSE* of 743.91 kg/ha, and a *MAPE* of 14.4% (Table 3). This was mainly because the multi-temporal remote-sensing reflectance could directly catch the crop's growth trajectory, simultaneously capturing the evolution of the aboveground foliage and providing essential information for the final grain yield [23]. The weather and soil property data decreased the accuracy compared to the remote-sensing data. Two factors may explain this finding. On the one hand, many regions rely heavily on irrigation, especially in the North China Plain. Therefore, the impact of weather conditions (primarily the amount of precipitation) on winter wheat yield is weaker in these regions than in other areas where irrigation is not widely performed. On the other hand, soil property data only reflects the primary soil conditions and does not account for the widespread use of fertilizers. Fertilization could significantly change the soil nutrient structure, reducing the

correlation between soil properties and winter wheat yield. In addition, it is interesting that the soil data alone performed better than the weather data. This shows a strong spatial correlation between soil data and winter wheat yield, consistent with the actual production. It also reflects the rationality of the proposed model, which comprised soil data as an independent static branch. Figure 10 shows the variation in the *RMSE* of winter wheat yield prediction for each data source from 2001 to 2015. The results showed that the dual-stream deep-learning neural network model achieved the highest prediction accuracy in most years. In drought years (2001–2003), soil data alone could achieve a comparable and relatively high accuracy with the remote-sensing data alone, while the weather data alone had relatively low accuracy in most years. This suggests that soil data can have significantly improved model accuracy in drought conditions.

**Table 3.** Performance of each data source for winter wheat yield prediction. The new model developed in this study combined all three datasets, but the analysis in this table examined each dataset separately. All $R^2$ values were statistically significant ($p < 0.05$). *MAPE*, mean absolute percentage error; *RMSE*, root-mean-square error.

| Dataset | $R^2$ | *RMSE* (kg/ha) | *MAPE* (%) |
|---|---|---|---|
| Remote sensing | 0.73 | 743.91 | 14.43% |
| Weather | 0.67 | 832.11 | 15.93% |
| Soil | 0.69 | 807.19 | 15.13% |
| **Dual-stream deep-learning neural network model** | 0.79 | 650.21 | 12.37% |

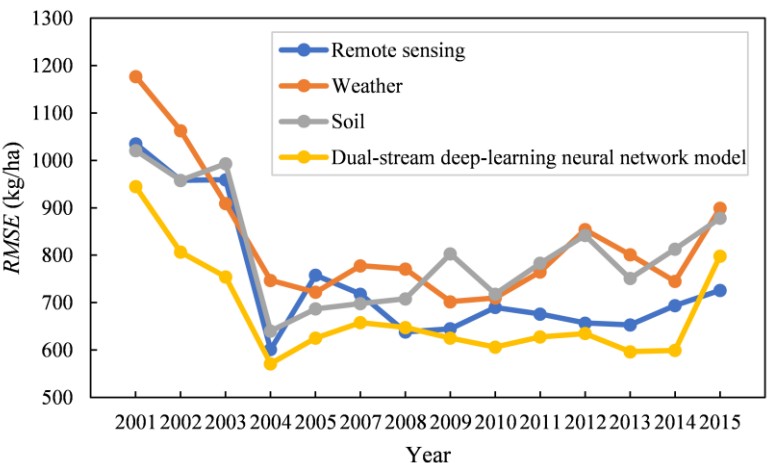

**Figure 10.** Root-mean-square errors (*RMSE*) of winter wheat yield prediction for each data source from 2001 to 2015.

All the factors in our analysis influenced the final crop yield (e.g., weather conditions, soil nutrients, and crop management practices). They would therefore have a more comprehensive impact on crop growth than any one group of factors, and the remote-sensing data could directly capture most of this effect. Therefore, it is logical that the remote-sensing data contributed more to the yield-prediction performance than the other datasets. Including weather and soil property data provided additional unique information that could further improve the prediction accuracy. This was consistent with several previous studies [18,57].

*5.2. Comparison with Traditional Methods*

To further evaluate the proposed yield prediction method's performance, we performed experiments to compare our results with previous methods such as a multiple regression, a support vector machine, and a random forest model (Table 4).

**Table 4.** Comparison of the present results with three traditional methods. All $R^2$ values were statistically significant ($p < 0.05$). *MAPE*, mean absolute percentage error; *RMSE*, root-mean-square error.

| Method | $R^2$ | *RMSE* (kg/ha) | *MAPE* (%) |
|---|---|---|---|
| Multiple regression | 0.55 | 971.04 | 19.07% |
| Random forest | 0.62 | 890.85 | 17.67% |
| Support vector machine | 0.55 | 966.87 | 19.66% |
| **Dual-stream deep-learning neural network model** | 0.79 | 650.21 | 12.37% |

The new dual-stream deep-learning neural network outperformed all three traditional methods. The multiple regression and support vector machines had similar performances ($R^2 = 0.55$), and although the random forest model had the highest accuracy ($R^2 = 0.62$) among the traditional methods, it had a much higher error (both *RMSE* and *MAPE*) and a lower $R^2$ than our new model. One reason may be that the predictor variables showed high collinearity and complexity. Therefore, these models could result in severe overfitting (i.e., fitting the training data well but achieving poor prediction results during validation). The random forest model was less sensitive to the collinearity of the input variables and therefore achieved better performance than multiple regression and a support vector machine. In contrast, deep learning could automatically extract, weigh, and fuse informative features from the input variables and was less affected by noise and collinearity in the input data, resulting in the highest performance.

*5.3. Comparison with Other Deep-Learning Yield-Prediction Methods*

Our results indicated that LSTM and GRU performed better in severe drought conditions. One possible reason is that recurrent neural networks have a higher ability to model short- and long-range dependencies between sequential data, leading to a more robust performance than convolutional neural network models. However, VGG had the worst performance among the seven deep-learning models we discussed. This was mainly because VGG has a less sophisticated structure than the others, which limited its ability to extract representative and robust features of the data. ResNet and DenseNet both adopt a residual connection structure, which has the merits of removing hierarchical features and improving gradient flow simultaneously, thereby improving performance compared with VGG. Unlike the other convolutional neural networks, the Inception-based model used several parallel convolutions at different scales, which improved its ability to extract multi-scale features. However, all convolutional neural networks risk overestimating the yield during years with abnormal conditions. Therefore, if extreme weather conditions exist during the training data periods, recurrent neural networks (especially LSTM) better account for those outlier years; under other circumstances, the Inception-based structure is recommended for yield prediction due to its ability to extract more relevant multi-scale features.

One difference between the present study and the previous studies in the USA relates to land use patterns. Land parcels in the U.S. and many other developed countries tend to be larger than the land parcels in China, with a single planting pattern. Therefore, mixed pixels create fewer problems affecting the model's performance. China has more fragmented land parcels, and the planting patterns constantly change, resulting in more mixed pixels than in the U.S. This makes accurate crop-yield prediction more challenging in China. The new model proposed in the present study achieved good accuracy, with *MAPE* = 12.4%, comparable to or better than previous state-of-the-art methods. The model performance would be further improved if remote-sensing data with finer spatial resolution became available, reducing the effect of mixed pixels.

### 5.4. In-Season Winter Wheat Yield Prediction

Winter wheat is often planted in September or October and harvested in May or June of the following year in China. Early and accurate prediction of winter wheat yield over large areas ahead of the harvest is essential for food security and planning import and export levels. To account for this, we explored how our model might be used for in-season yield prediction in China's major winter wheat planting area (Figure 11).

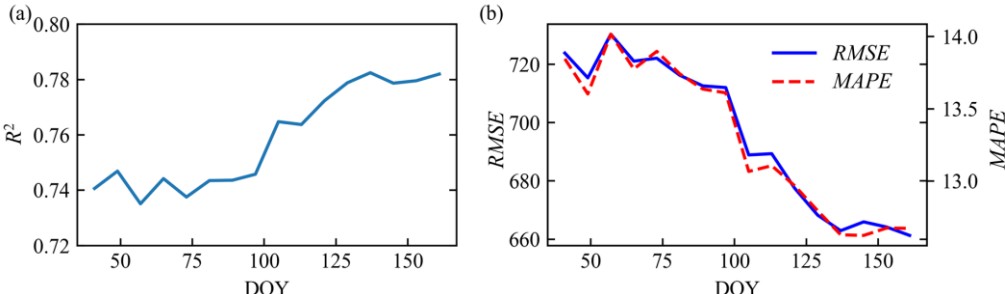

**Figure 11.** Model performance as a function of the day of year (DOY): (**a**) $R^2$ and (**b**) the mean absolute percentage error (*MAPE*) and root-mean-square error (*RMSE*). The results represent means from 2001 to 2015.

Considering that the period after green-up is most important for winter wheat yield [22], we utilized soil property data and only the subset of the remote-sensing and weather data during the green-up period to train and test the yield-prediction model. Figure 11 shows the results of predicting the final yield before harvest based only on this early-season data up to the point at which the prediction is calculated. The model performance was weakest between DOY 41 and DOY 97 (green-up to the booting period), with some fluctuations, and the mean $R^2$ was 0.74. This was probably due to the insufficient information conveyed by the limited remote-sensing and weather data, which limited our model's ability to characterize the growth status of winter wheat. As the amount of sequential data available increased, the prediction performance improved after DOY 97 (booting period), reaching a maximum $R^2$ of 0.78 at both DOY 137 (heading to flowering) and DOY 161 (from flowering to maturity). Figure 11 also shows that $R^2$ increased rapidly between DOY 97 and DOY 105 (the heading stage) and that both *RMSE* and *MAPE* decreased, indicating that the proposed model could obtain a reliable yield-prediction result about two months before the winter wheat harvest.

### 5.5. Possibility of Establishing a Parcel-Level Crop Yield-Prediction Model

With the development of precision agriculture, accurate parcel-level or field-level crop yield information is in great demand because it would reveal crop growth and yield responses to field management practices and environmental stress [75–77]. However, compared with county-level yield prediction, for which remote-sensing data with coarse resolution (e.g., MODIS) would meet the data requirements for the analysis, parcel-level prediction needs data with finer spatial resolution, such as Landsat-8 and Sentinel-2 data or even the harmonized Landsat-8 and Sentinel-2 data to avoid interference from cloud cover.

Researchers have also proposed several parcel-level crop yield predictions [75–77]. However, these methods were mainly based on simple regression models, whose accuracy and robustness could be further improved by considering a deep-learning approach. The new model we developed could be easily extended to parcel-level crop yield prediction if parcel-level datasets become available, including crop yield data, high-resolution remote-sensing data, and (based on the benefits demonstrated in this paper) soil data. Other factors that may influence the crop yield at the parcel level should also be accounted for, such as winter wheat varieties and field-management measures (e.g., fertilization and irrigation).

Any parcel-level yield prediction based on deep learning would also require a large quantity of site-specific yield data. However, collecting a large number of representative parcel-level yield measured data to support such an analysis is time-consuming and remains challenging. To address this issue, high accuracy, with less uncertainty of parcel yield from data assimilation, should be considered in the future to provide ground-truthing data for training and testing of the deep-learning models [6].

### 5.6. Limitations and Future Perspective

It is essential to discuss the uncertainty of our new model to show its limitations, as this discussion could provide vital clues for future studies. Specifically, the model uncertainty could be analyzed from the data and the model levels.

As for the input variables of the deep-learning model, we only included remote-sensing, weather, and soil property data from the data perspective. Other features were not considered, which are also closely related to final crop yield (e.g., soil moisture content, winter wheat varieties, irrigation, and fertilization). Among these data sources, soil moisture content during the growing season is critical for determining the final crop yield [18]. Therefore, future research could introduce remote-sensing data obtained by passive or active microwave sensors to retrieve soil moisture content and provide complementary information that would improve crop yield prediction. Moreover, due to a lack of data under extreme weather conditions, the proposed deep-learning model is less able to accurately predict crop yield responses to extreme climate change, as shown by the decreased accuracy during a drought period (i.e., the arid period from 2001 to 2003). With ongoing data collection to support model training, the deep-learning model would likely become easier to generalize, resulting in a better yield prediction under various meteorological disaster scenarios.

The values of the predictor variables for each county were generated by averaging all the pixel values within the county's borders, which means that the spatial variation in the input data (especially the remote-sensing data) was not accounted for, leading to a loss of useful information. Therefore, future studies should consider more sophisticated spatial feature extraction methods. From this perspective, two-dimensional convolutional neural networks might be introduced instead of the approach we used due to the superior ability of two-dimensional convolution to extract spatial features from the original pixels. The cropland cover data in this study was obtained from the MODIS land-cover-type product, which does not differentiate among crop types. Although previous studies [52,67] justified the use of this product by noting that it did not significantly decrease the yield-prediction performance, future research should use a winter wheat mask with a finer spatial resolution to reduce estimation errors.

From the modeling perspective, we only considered a few recent convolutional neural networks and recurrent neural network models. More structures could be examined to explore their potential, such as CapsuleNet [78]. In addition, because each growth stage of a crop has a different impact on the final yield, it will be essential to identify and account for the key stages and their interactions with factors such as temperature and precipitation. Hence, a future study could introduce an attention mechanism [79,80] into the yield-prediction model, allowing the model to learn the explicit representations of crop growth cycles to increase the interpretability of the deep-learning models.

Wheat cultivation in China occurs against a background of high variation in winter wheat's phenological stages, which remains challenging for accurate yield prediction. Despite those challenges, the proposed deep-learning model still obtained promising results, with high suitability for large-scale crop-yield prediction. However, prior studies [81,82] showed that splitting a large study area into smaller agro-climatic zones and establishing a prediction model for each zone could improve the model's performance. Hence, a future study could also consider building separate yield-prediction models for each agro-climatic zone.

Given that attention mechanisms have been widely used in many remote-sensing classification applications [83–85], this method could help to interpret how the deep-learning model makes its decisions. For wheat yield estimation, attention could be added to a recurrent network (e.g., LSTM) to give more information about the importance of each period of the winter wheat's growth stages. Furthermore, a bi-directional attention structure might be a better choice due to its ability to model the dependencies between two periods in a forward–backward manner, revealing more clues about which period contributes most to the yield estimation. For instance, with the help of an attention mechanism, the deep-learning model could tell that if the temperature is too high during a specific period, the yield will decrease by a predictable amount. Therefore, further research is justified in using an attention mechanism, especially bi-directional attention, to improve the interpretation ability of deep-learning models in yield estimation.

## 6. Conclusions

In this study, a novel deep-learning model was developed for winter wheat yield prediction based on multi-source data (remote-sensing, weather, and soil properties) in China's major winter wheat planting areas. We proposed a dual-stream deep-learning neural network based on well-known, previously developed deep-learning structures (VGG, ResNet, DenseNet, Inception, LSTM, and GRU) with non-linear relationships between winter wheat yield and the multi-source predictor indicators. We found that the Inception-based model achieved the highest accuracy, with an $R^2$ of 0.79, an *RMSE* of 650.21 kg/ha, and a *MAPE* of 12.4%. The prediction model's errors were randomly distributed and showed no obvious under- or over-estimation.

We also investigated the influence of the predictor variable on winter wheat yield prediction. The remote-sensing data had a stronger contribution in predicting yields than the weather and soil property data. One possible explanation is that the remotely sensed surface reflectance provides a more comprehensive representation of the crop's growth and stress situations. The dual-stream deep-learning neural network model outperformed the traditional multivariate regression and machine-learning models by increasing the $R^2$ by 17.0 to 24.1% and decreasing *RMSE* by 240.6 to 320.8 kg/ha. The dual-stream deep-learning neural network model could also provide an accurate yield prediction about two months before harvesting, with predicted values within 13% of the official reported yield statistics, showing an excellent in-season prediction capability.

Our study demonstrated the advantages of using a deep-learning model for the large-scale, long-term prediction of winter wheat yields across China at a county level. The proposed method, without particular constraints on crop type or region, would be likely to generalize to other crops (e.g., maize, paddy rice) and other similar agricultural planting regions worldwide with minimal effort.

**Author Contributions:** Conceptualization, J.H. and H.H.; methodology, H.H.; software, X.W.; validation, Q.N., Q.F. and J.L.; formal analysis, H.H.; investigation, Q.N.; resources, J.H.; data curation, J.L.; writing—original draft preparation, H.H.; writing—review and editing, J.H. and X.L.; visualization, Q.F. and X.W.; supervision, X.L.; project administration, J.H.; funding acquisition, J.H. All authors have read and agreed to the published version of the manuscript.

**Funding:** This research was funded by the National Natural Science Foundation of China (Project No. 41971383 and 42271339).

**Data Availability Statement:** Not applicable.

**Acknowledgments:** We would like to express our sincere thanks to the anonymous reviewers.

**Conflicts of Interest:** The authors declare no conflict of interest.

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
