# Peer review of "Developing a Dual-Stream Deep-Learning Neural Network Model for Improving County-Level Winter Wheat Yield Estimates in China"

_remotesensing, doi:10.3390/rs14205280_

Round 1

Reviewer 1 Report

Summary

This manuscript applies deep learning methods to predict county-average wheat yields in China. Input data for the models come from remote sensing data (MODIS), climate data and soil maps (SoilGrids250m). Several versions of deep learning models are compared, and these are also compared with other machine learning algorithms and with linear regression, showing the best performance from some of the deep learning models. I think that the manuscript provides a useful comparison of methods, but have some suggestions to improve its utility and clarity that I would recommend before publication.

General comments

It wasn’t clear to me what the purpose of prediction was. It seems to me that you have data as county-averaged yields, and want to predict also county-level yields; but are there some counties/years where there are missing data that you want to fill? Or is the purpose to be able to predict the county-averaged yields before the data for that season are released? (In this case, should you not test in a leave-one-year-out validation test, where all the data from a particular target year and omitted from a calibration set, a model fitted using that calibration set to predict the yields for the target year?)

Related to this, I see the benefits of early-season prediction (based on remote sensing and climate data up to a point in time before harvest), as addressed in the discussion section, but I was a bit unsure about validation details (did the calibration dataset omit all yield data from the season being forecasted?).

I was also a bit confused about how validation was done – were models and predictions done separately for each year? Or were all years included in a single model? I think the former, though the latter would be more useful.

Detailed comments

Section 2.2.5 : I couldn’t see how many data (counties) there were for each year

Line 200 : were all climate and soil maps also masked to the cropping area before averaging (I think they should be), or just the remote sensing data?

Line 202 : SOC would vary somewhat in time, for instance you would normally expect a decline in SOC when native vegetation was first converted to cropland (though less temporally variable than climate and remote sensing). Also might be worth mentioning that these are soil maps based on digital soil mapping, so would have considerable uncertainty at the point scale (grid), but perhaps less uncertainty as the county-averaged data that you use.

Eq 1 : this seems to be the sum of squared errors, not the mean squared error

Eq 2-4 : subscripts and superscripts are inconsistent and need sorting out

Line 316 and later : You mention ‘over estimated’ and later talk about ‘unbiased’ predictinos, so I think it would be worth including the mean error in your validation statistics (Table 2) to back this up.

Table 3 : Just a comment, but I’m surprised how well soil on its own did (R2 = 0.69); the results in this table suggest to me that perhaps a lot of the information is coming from spatial auto-correlation that is not directly included in the model, but is indirectly included through the spatial auto-correlation of the covariates.

Figure 9 – are these predictions from a model fitted with all of the data, or from a calibration subset?

Reviewer 2 Report

This manuscript focused on county-level winter wheat yield estimates with deep learning method and multi-source data. Overall, the organization and structure of the manuscript are appropriate and well structured, and the topic is relevant to the scope of the journal. I personally recommended the minor revisions of this manuscript as some small issues needed to be handled.

1. Line 21: “achieved an average R2 …”, the squared term should be superscript.

2. Line 104: There is an extra space between the words "have" and "tried"

3. Line 123-124, Line 127-128: This information is not shown in Figure 1, the author needs to modify the figure or the text.

4. As shown in Table 3, The remote sensing data can get acceptable performance, what is the role of soil and weather data? Especially, soil and wheat yield are closely related in actual production. Can some discussion on the effect of soil texture and soil heterogeneity on wheat yield prediction be added to highlight the advantages of the proposed model in this aspect of soil factors?

5. The performance of each data source for winter wheat yield prediction were analyzed in Section 5.1, how about their accuracy under different conditions? As the author mentioned, years of 2001 to 2003 suffered severe drought in China. Will these data have different impacts with or without such stressful conditions?

Reviewer 3 Report

The Authors designed a model to forecast winter wheat yields over a large spatial area
in major cultivation regions in China.
They used detection, weather and soil properties as well as neural networks for modeling.
The work is well written and fits within the scope of the journal.

The work is generally correct, as well as the network and training.
The Authors have a lot of data and fortunately the data are for 14 years.
They showed an error, they compared their results with other methods.
The results are generally well described, e.g. a significantly larger prediction error in 2001-2003.
Picture 9c is interesting: it can be seen that the greatest prediction error
is in the parts of the country where there is the least data.
It could be check to see if by any chance there is some strange geography (mountains, lakes, etc)
that would increase the variance.

A table would be useful: the number of training vectors, their size,
the number of trainable parameters for each branch of their network.

It would be good to have some basic information about the
administrative units in the article.

The Authors could give the wheat growing cycle and what the temperature then. Is in all
regions a similar cycle of plant growth and development?
It could be a problem because in one analysis, a different development phases are compared.

Table 1 lists the variables. As for satellite data, the Authors used individual channels (light ranges) for the analysis,
not vegetation indices based on these channel to avoid the effects of different radiation
the sun at different times of the year.

The Authors used SoilGrids where soil pH is constant which is not always the case.
